# Deep Autoregressive Models as Causal Inference Engines

**Daniel Jiwoong Im**                                                      *ji641@nyu.edu*
*Center for Data Science*
*New York University*

**Kevin Zhang**                                                      *kyz2005@columbia.edu*
*Computer Science*
*Columbia University*

**Nakul Verma**                                                      *verma@cs.columbia.edu*
*Computer Science*
*Columbia University*

**Kyunghyun Cho**                                                      *kyunghyun.cho@nyu.edu*
*Center for Data Science*
*New York University*

**Reviewed on OpenReview:** *https://openreview.net/forum?id=uuREHPf2ll*

## Abstract

Existing causal inference (CI) models are often restricted to data with low-dimensional confounders and singleton actions. We propose an autoregressive (AR) CI framework capable of handling complex confounders and sequential actions commonly found in modern applications. Our approach accomplishes this using *sequencification*, which transforms data from an underlying causal diagram into a sequence of tokens. Sequencification not only accommodates training with data generated from a large class of DAGs, but also extends existing CI capabilities to estimate multiple causal quantities using a *single* model. We can directly compute probabilities from interventional distributions, simplifying inference and improving outcome prediction accuracy. We demonstrate that an AR model adapted for CI is efficient and effective in various complex applications such as navigating mazes, playing chess endgames, and evaluating the impact of certain keywords on paper acceptance rates, where we consider causal queries beyond standard reinforcement learning-type questions.

## 1 Introduction

Modeling causal relationships is important across various fields for informed decision-making (Holland, 1986; Cochran & Rubin, 1973; Pearl, 2010). However, existing causal inference (CI) algorithms are often limited by their inability to handle sequential variable-length actions and high-dimensional covariates (Louizos et al., 2017; Kumor et al., 2021; Im et al., 2021; Lu et al., 2022; Zhong et al., 2022). This work aims to address these shortcomings by designing a CI engine applicable to complex data involving high-dimensional variables.

Consider a medical setting where a doctor prescribes a sequence of treatments to the patient. In this scenario, the number of treatments that the patient undergoes is not necessarily fixed (e.g. additional procedures may be assigned depending on patient response or two medications might be combined into one). As a result, the number of possible treatment sequences grows combinatorially. Moreover, the medical decision may be dependent on high-dimensional data describing the condition of the patient, such as text from electronic health records. For successful treatment effect estimation, we want a framework that can accommodate arbitrary-length action sequences for complex, high-dimensional confounding data. To tackle this problem, we propose using neural network-based autoregressive (AR) models for causal inference.

Autoregressive (AR) models are a standard framework for learning conditional probability distributions and predicting values in sequential or time-series data. Neural network-based AR models are widely used in applications such as language modeling for next-token prediction and text generation (Devlin et al., 2019; Radford et al., 2019; Touvron et al., 2023). As demonstrated by large language models (LLMs), AR models can capture complex relationships and scale effectively to large datasets. Recent studies (Gupta et al., 2023; Zhang et al., 2024; Xu et al., 2024) show that fine-tuning pre-trained LLMs utilizes knowledge from an internet-scale text corpus, enhancing performance on various tasks.

We demonstrate that AR models can also be employed for causal inference by transforming observational data into a sequence following an underlying causal ordering. To achieve this, we employ *sequencification* for representing data based on a known causal diagram, specified as a directed acyclic graph (DAG). A neural network-based AR model can then learn the conditional probability distributions implied by the DAG. This allows for efficient sampling of sequential actions and high-dimensional confounders, enabling Monte Carlo estimation to approximate various causal effects. Furthermore, since the AR model learns all conditional distributions in the sequence, a *single* model trained on sequencified data can be used to compute a wide range of interventional distributions.

Our methodology is designed to handle variable-length sequential actions, combinatorially large action spaces, and high-dimensional confounders. These capabilities encompass a variety of common causal inference tasks, including average treatment effect (ATE) estimation, individual treatment effect (ITE) estimation, interventional distribution approximation, etc. To the best of our knowledge, all prior work can only accommodate fixed-length covariates and actions, and most evaluate on relatively low-dimensional data.

We conduct empirical studies across a variety of exemplar tasks, such as navigating mazes, playing chess endgames, and evaluating the impact of specific keywords on paper acceptance rates. The experiments show that our framework can (1) infer causal effects involving high-dimensional variables, (2) generalize to unseen confounders and action sequences, and (3) leverage pre-trained LLMs to answer text-based causal questions.

## 2 Related work

Previous work has explored the use of deep learning or AR models for estimating causal effects. Here, we provide a brief overview of the relevant studies.

**Sequencification for statistical engines.** Various machine learning fields have used linearized representations for tasks. In natural language processing (NLP), linearization (which we refer to as sequencification) is used to convert a syntactic tree into a sequence for building language model-based parsers (Vinyals et al., 2015; Liu et al., 2022; Sheng et al., 2023). In reinforcement learning (RL), an episode can be encoded as a sequence of states, actions, and rewards. An autoregressive model is then trained on these sequences to capture relationships among the variables (Chen et al., 2021; Janner et al., 2021). These instances suggest that AR models trained on sequencified data can effectively learn statistical dependencies among multiple high-dimensional variables.

**Language models as causal engines.** Several works have used language models for high-dimensional CI tasks (Feder et al., 2021; Egami et al., 2022; Veitch et al., 2020). A key challenge across these studies is satisfying positivity constraints (D'Amour et al., 2021; Tu & Li, 2022; Gui & Veitch, 2023). Techniques from NLP, such as topic models (Sridhar & Getoor, 2019; Mozer et al., 2020), latent variable models (Keith et al., 2020), and contextual embeddings (Veitch et al., 2020), have been used to produce low-dimensional embeddings that can satisfy positivity. In addition to using NLP techniques to reduce the dimensionality of observations, natural language can also serve as a proxy for observed confounders. For example, Roberts et al. (2020) apply a text-matching algorithm using contextual embeddings and topic models to estimate causal effects based on proxy texts.

**Deep learning for causal engines.** Various deep neural network architectures have been proposed for CI. Representation learning for CI often incorporates a regularization term that enhances generalization for counterfactual actions (Shalit et al., 2016; Johansson et al., 2018; Wang & Jordan, 2024). Deep latent variable CI models learn stochastic latent variables to model potential outcomes using a richer family of

distributions (Louizos et al., 2017; Kocaoglu et al., 2017; Im et al., 2021). Normalizing flows have also been used for causal effect estimation. For example, Krause et al. (2023) develop a doubly robust density estimator for potential outcomes by combining a nuisance and target normalizing flow.

For deep autoregressive models, Monti et al. (2020) introduce an AR flow model that learns an invertible density transformation between variables. Their approach enables direct computation of interventional and counterfactual distributions without the need for complex latent variable manipulations. Garrido et al. (2021) use neural AR density estimators (Larochelle & Murray, 2011) to model causal mechanisms and predict causal effects using Pearl's do-calculus (Pearl, 2009).

Other studies examine treatment effect estimation in more complex, multi-dimensional scenarios. Frauen et al. (2025) propose a model-agnostic learner for a sequence of actions over a fixed time horizon. An interesting line of work focuses on real-valued actions (e.g. amount of medical dosage administered) for heterogeneous dose-response estimation using different neural-network architectures, including generative adversarial networks Bica et al. (2020), varying coefficient models Nie et al. (2021), and contrastive representation learning Zhu et al. (2024).

**Causal Generative Models.**  Another approach to CI models data as part of a generative process and learns a causal generative model. These methods typically parameterize relationships in a known causal diagram using neural networks. For causal effect identification and estimation, Xia et al. (2023) use neural causal models trained via gradient-based optimization with a minimization-maximization objective. Rahman & Kocaoglu (2024) introduce a modular learning framework for optimizing causal generative models in semi-Markovian settings by decomposing the data distribution into $c$-factors (Tian & Pearl, 2002). Their method can use pre-trained generative models to learn individual conditional distribution components.

**Existing limitations.**  Previous works exhibit notable weaknesses compared to our approach. First, most methods are validated only on relatively low-dimensional variables (tens of dimensions) and fixed-length actions. In contrast, our AR model is designed to handle high-dimensional confounders and variable-length series of actions where the number of possible sequences grows exponentially. This makes it applicable to settings with high-dimensional data and arbitrarily long sequence of treatments. Second, prior studies often focus on estimating specific causal effects or learning generative models by optimizing individual components of the causal structure. We instead use a *unified* end-to-end AR model that can estimate multiple causal queries. Third, we extend existing CI capabilities by leveraging pre-trained language models for settings where domain knowledge is essential for accurate inference (e.g. NLP).

## 3 Background

This section outlines the background knowledge of causal effect estimation and language models necessary for understanding our methodology.

### 3.1 CI problem formulation

We study interactions between the following set of variables: an observable confounder $X$, action $A$, and outcome $Y$. Causal relationships are represented as a directed acyclic graph (DAG), where edges denote direct effects (cf. Figure 1a). By applying the backdoor adjustment formula (Pearl, 2009), we can compute the potential outcome resulting from an intervention on the action:

$$Y_a := \mathbb{E}_Y[Y \mid \mathsf{do}(A = a)] := \sum_{x,y} y \cdot p(Y = y \mid A = a, X = x)p(X = x). \tag{1}$$

The notation $\mathsf{do}(A = a)$ represents an intervention on $A$, setting its value to $a$. Typically, the confounder $X$ is assumed to be low-dimensional to avoid computing density estimates $p(x)$ in high dimensions. Our goal is to model causal effects when typical assumptions in prior CI work are violated, including settings with complex confounders and combinatorially large action spaces.

We make the following set of standard assumptions in causal inference: (1) unconfoundedness, which states that $Y_0, Y_1 \perp\!\!\!\perp A \mid X$, and (2) positivity, which requires that $p(a, x, y) > 0$ for all triplets $(a, x, y)$.

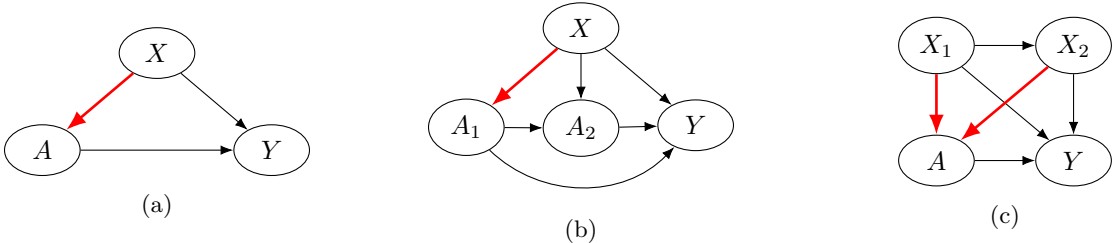

Figure 1: Causal diagrams illustrating interventions on the action (a), partial action (b), and when conditioning on a prefix of the confounders (c), where $X$, $A$, and $Y$ denote the confounders, actions, and outcome respectively. Bold red arrows indicate the pathways blocked by the corresponding intervention. Potential outcomes can be computed using the backdoor adjustment formula for each scenario.

## 3.2 Language models

Language models (LMs) are designed to predict and generate text by learning linguistic patterns from a training corpus. Let $\mathbb{W}$ denote the vocabulary of the text corpus, which includes special $\langle \mathtt{start} \rangle$ and $\langle \mathtt{end} \rangle$ tokens. LMs estimate the probability of a sequence of tokens $\mathbf{w} = (w_1, \cdots, w_T) \in \mathbb{W}^T$ in an autoregressive manner, where $w_1 = \langle \mathtt{start} \rangle$ and $w_T = \langle \mathtt{end} \rangle$. The probability of the sequence $\mathbf{w}$ is decomposed into the product of next-token probabilities: $p(\mathbf{w}) = p(w_1) \cdot p(w_2 \mid w_1)p(w_3 \mid w_1, w_2) \cdots p(w_T \mid w_1, \ldots, w_{T-1})$.

# 4 Language models as statistical inference engines

As suggested by Equation 1, CI requires accurate estimation of statistical quantities to calculate causal effects. In this section, we describe how an AR model can be adapted into a statistical inference engine for any DAG involving a set of confounders, actions, and outcomes using sequencification.

## 4.1 Causal graphs

We assume the underlying causal DAG $\mathcal{G} = (\mathcal{V}, \mathcal{E})$ is known, where the vertices $V_i \in \mathcal{V}$ represent random variables and the edges $E_{i \to j} \in \mathcal{E}$ denote conditional dependencies. Moreover, the graph satisfies the Markov property, meaning the joint probability distribution can be factored into a causally-consistent ordering,

$$p_{\mathcal{G}}(V_1, V_2, \ldots, V_M) = \prod_{i=1}^{M} p_{\mathcal{G}}(V_i \mid \mathrm{Pa}(V_i)),$$

where $\mathrm{Pa}(V_i) \coloneqq \{V_j \in \mathcal{V} \mid E_{j \to i} \in \mathcal{E}\}$ is the set of parent nodes of $V_i$. We assume that the graph $\mathcal{G}$ is fully specified and that all variables within $\mathcal{G}$ are observed.

## 4.2 Sequencification

Suppose $V_i$ takes the value $\mathbf{v}_i$ from its corresponding distribution. Let string$(\cdot)$ be an injective function that maps $\mathbf{v}_i$ to a sequence of tokens: string$(\mathbf{v}_i) = (\langle \mathtt{start}_i \rangle, w_1, w_2, \ldots, w_{L_{\mathbf{v}_i}})$. Here, $\langle \mathtt{start}_i \rangle$ is a special token indicating the beginning of the string representation for $\mathbf{v}_i$, and $L_{\mathbf{v}_i}$ is the length of string$(\mathbf{v}_i)$ excluding the $\langle \mathtt{start}_i \rangle$ token. We define $\langle \mathtt{start}_i \rangle$ uniquely for each $i$ so that each random variable can be uniquely identified from its string representation by its corresponding initial token.

Let $\mathbf{t} = (V_{i_1}, V_{i_2}, \ldots, V_{i_M})$ be a permutation of the random variables. We say $\mathbf{t}$ is a *topological ordering* if $V_i$ precedes $V_j$ in the ordering for all edges $E_{i \to j}$. Let $\mathcal{T}$ denote the set of all topological orderings. Consider $N$ samples drawn from the underlying causal diagram $\mathcal{G}$: $(\mathbf{v}_1^{(n)}, \mathbf{v}_2^{(n)}, \cdots, \mathbf{v}_M^{(n)}) \sim p_{\mathcal{G}}(V_1, V_2, \cdots, V_M)$ for $n = 1, \ldots, N$. For each sample, we construct a string $\mathbf{s}^{(n)}$ by concatenating the string representations of all random variables according to a topological ordering $\mathbf{t}^{(n)}$ selected uniformly at random from $\mathcal{T}$.

Formally, $\mathbf{t}^{(1)}, \mathbf{t}^{(2)}, \ldots, \mathbf{t}^{(N)} \overset{i.i.d.}{\sim} \text{Uniform}(\mathcal{T})$ and

$$\mathbf{s}^{(n)} = \text{string}(\mathbf{v}_{i_1}^{(n)}) \oplus \text{string}(\mathbf{v}_{i_2}^{(n)}) \oplus \cdots \oplus \text{string}(\mathbf{v}_{i_M}^{(n)}) \oplus \langle \text{end} \rangle,$$

where $\mathbf{t}^{(n)} = (V_{i_1}, V_{i_2}, \ldots, V_{i_M})$ and $\oplus$ denotes string concatenation. We refer to the process of converting observed samples into a sequence of tokens as *sequencification.*

Sequencification supports any data that can be transformed into a linear sequence of tokens. For example, tokens may represent specific values for numerical (e.g., binary) data or subwords for text, depending on the problem domain. Our approach can also handle mixed data modalities, as different variables may have separate tokenization strategies.

### 4.3 Randomized topological orderings

We randomize over topological orderings consistent with the causal graph to obtain robust estimates of conditional probabilities. For instance, if a node in the graph has multiple independent parents, randomizing the order in which the parents are sequenced helps prevent the model from overfitting to any particular ordering. As a result, different samples may be concatenated using distinct topological orderings. However, because the data is generated by ancestor sampling $\mathbf{v}_i^{(n)} \mid \text{Pa}(\mathbf{v}_i^{(n)})$, values that causally influence $\text{string}(\mathbf{v}_i^{(n)})$ will always appear earlier in $\mathbf{s}^{(n)}$.

A natural question that may arise is how to obtain a good estimate when samples are sequencified according to different topological orderings. This is achieved by using the special $\langle \text{start}_i \rangle$ token at the start of each variable $V_i$ during sequencification, which indicates its position in the sequence (regardless of the topological ordering used).

### 4.4 Autoregressive statistical inference engines

After sequentification, we can train an AR language model, parameterized by $\theta$, on the sequencified dataset $D = \{\mathbf{s}^{(1)}, \mathbf{s}^{(2)}, \ldots, \mathbf{s}^{(N)}\}$ by the minimizing negative log-likelihood:

$$\mathcal{L}(\theta) = -\frac{1}{N} \sum_{n=1}^{N} \log p_\theta(\mathbf{s}^{(n)}) = -\frac{1}{N} \sum_{n=1}^{N} \sum_{t=1}^{|\mathbf{s}^{(n)}|} \log p_\theta \left( \mathbf{s}_t^{(n)} \mid \mathbf{s}_{1:t-1}^{(n)} \right), \tag{2}$$

where $|\cdot|$ denotes the length of a string and $\mathbf{s}_t^{(n)}$ is the $t^{\text{th}}$ token in $\mathbf{s}^{(n)}$. The trained model can estimate any conditional probability on $\mathcal{G}$ by computing $p_{\mathcal{G}}(V_i \mid \text{Pa}(V_i)) \simeq p_\theta(\mathbf{v}_i \mid \text{Pa}(\mathbf{v}_i))$. This is efficiently done by autoregressively traversing the sequence and calculating next-token probabilities using Monte Carlo estimation over the topological orderings.

## 5 Language models as causal inference engines

In this section, we illustrate how to infer causal effects by leveraging statistical quantities from a trained AR model, thereby transforming it into a causal inference engine. After learning the conditional distribution over $\mathcal{G}$ using an AR model, we can estimate causal quantities by deriving the appropriate identification formula from the known causal diagram. Sequencification, combined with knowledge of $\mathcal{G}$, allows us to estimate various causal effects.

### 5.1 Estimating causal quantities

We express the CI problem as a language modeling task. Given our DAG that consists of three variables (the observed confounder $X$, action $A$, and outcome $Y$), we sequencify the data as follows:

$$\mathbf{s}^{(n)} = \text{string}(\mathbf{x}^{(n)}) \oplus \text{string}(\mathbf{a}^{(n)}) \oplus \text{string}(y^{(n)}) \oplus \langle \text{end} \rangle.$$

In our formulation, $\mathbf{x}^{(n)}$ and $\mathbf{a}^{(n)}$ can be high-dimensional vector values and the action space can be combinatorially large. Without loss of generality, we treat the outcome variable $Y$ as a scalar, represented using a single token.

We can use the trained AR model to compute the distribution of $Y$ after intervening on $A$. This is typically intractable when $X$ is high-dimensional because the sample complexity of computing a non-parametric density estimate is exponential in the number of dimensions. However, we can approximate the interventional distribution by sampling from $p_\theta(X)$ and applying Monte Carlo estimation.

$$p_\theta(Y = y \mid \mathsf{do}(A = \mathbf{a})) = \sum_{\mathbf{x}} p_\theta(y \mid A = \mathbf{a}, \mathbf{x}) p_\theta(\mathbf{x}) \simeq \frac{1}{S} \sum_{s=1}^{S} p_\theta(y \mid A = \mathbf{a}, \boldsymbol{x}^{(s)}), \tag{3}$$

where $\boldsymbol{x}^{(s)} \sim p(X)$.

Furthermore, we can intervene on a prefix subsequence of $A$ even when the action space is large. By expressing $A = A_1 \oplus A_2$ and marginalizing out $A_2$, we can compute the effect of intervening on only $A_1$.

$$p(Y = y \mid \mathsf{do}(A_1 = \mathbf{a}_1)) = \sum_{\mathbf{x}} \sum_{\mathbf{a}_2} p_\theta(y \mid A_1 = \mathbf{a}_1, A_2 = \mathbf{a}_2, \mathbf{x}) p_\theta(\mathbf{a}_2 \mid \mathbf{a}_1, \mathbf{x}) \, p_\theta(\mathbf{x}) \tag{4}$$

$$\simeq \frac{1}{S} \sum_{s=1}^{S} p_\theta(y \mid \mathbf{a}_1, \boldsymbol{a}_2^{(s)}, \boldsymbol{x}^{(s)}), \tag{5}$$

where $\boldsymbol{x}^{(s)} \sim p(X)$ and $\boldsymbol{a}_2^{(s)} \sim p(A_2 \mid \mathbf{a}_1, \boldsymbol{x}^{(s)})$. For combinatorially large action spaces, Equation 4 is generally intractable because the marginalization requires exponentially many operations.

Similarly, we can condition on a prefix of the confounder by letting $X = X_1 \oplus X_2$ and marginalizing out $X_2$.

$$p(Y = y \mid \mathsf{do}(A = \mathbf{a}), \mathbf{x}_1) = \sum_{\mathbf{x}_2}^{X_2} p_\theta(y \mid A = \mathbf{a}, \mathbf{x}_2, \mathbf{x}_1) \, p_\theta(\mathbf{x}_2 \mid \mathbf{x}_1) \simeq \frac{1}{S} \sum_{s=1}^{S} p_\theta(y \mid A = \mathbf{a}, \boldsymbol{x}_2^{(s)}, \mathbf{x}_1) \tag{6}$$

where $\boldsymbol{x}_2^{(s)} \sim p(X_2 \mid X_1 = \mathbf{x}_1)$. The causal diagrams for these scenarios are shown in Figure 1. Note that we can only intervene on prefixes of the action $A_1$ (or condition on prefixes of the confounder $X_1$) because this ensures the marginalization step samples $A_2$ (or respectively $X_2$) conditioned on preceding variables.

We emphasize that all causal quantities can be computed by a *single* language model trained on sequencified observations. Our approach enables efficient sampling and computation of conditional $p(Y \mid A = \mathbf{a})$, interventional $p(Y \mid \mathsf{do}(A = \mathbf{a}))$, partial interventional $p(Y \mid \mathsf{do}(A = \mathbf{a}_1))$, and conditional interventional $p(Y \mid \mathsf{do}(A = \mathbf{a}), X_1 = \mathbf{x}_1)$ distributions, all using a unified model. This provides an end-to-end framework for computing a variety of causal queries using a single model.

While the partial interventional distribution $p(Y \mid \mathsf{do}(A_1 = a_1))$ is computable by effectively discarding $A_2$, training an AR model on sequencified data that explicitly includes $A_1$ and $A_2$ is more flexible. Our approach can efficiently estimate not only partial interventional distributions but also interventions on $A_1$ and $A_2$ simultaneously, providing a unified framework for handling multiple causal queries. A similar argument holds for partially conditioning on the confounders.

## 6 Experiments

We demonstrate the effectiveness of our approach in estimating causal effects for sequential actions and high-dimensional confounders while also assessing robustness to distribution shifts. Our experiments showcase the ability to (1) infer potential outcomes with sequential actions and high-dimensional confounders, (2) efficiently approximate potential outcomes via Monte Carlo sampling, and (3) leverage knowledge from a pre-trained LLM. We evaluate our method across three environments: a maze setting for navigational decision-making, a chess environment analyzing strategic moves in king vs. king-rook endgames, and the `PeerRead` dataset which examines the impact of theorem presence on academic paper acceptance.

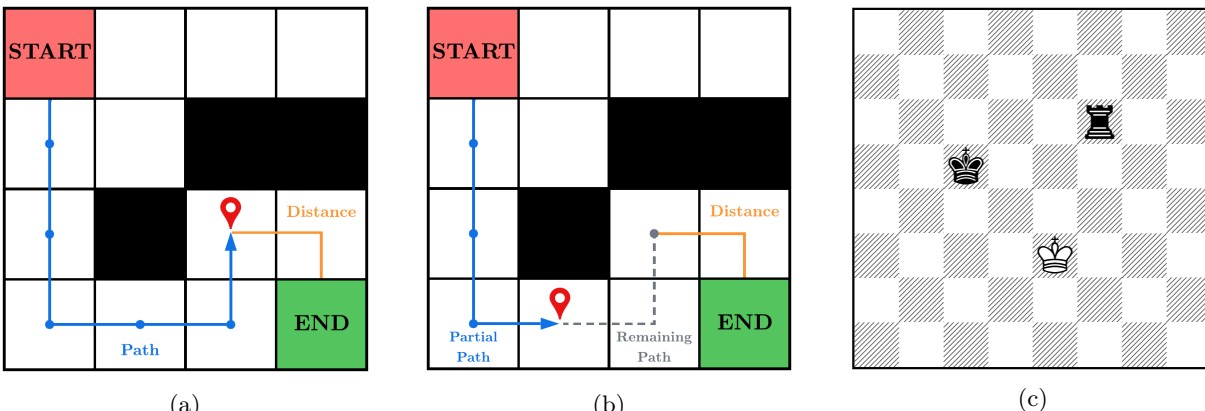

Figure 2: Illustrations of the maze and chess experimental settings. In the maze experiment, we address two questions: what is the potential outcome given (a) a complete path, and (b) a partial path? The blue path represents the intervention, gray indicates a potential remaining path after a partial intervention, and orange denotes the distance to the end. The chess experiment aims to determine which pieces Black should move to checkmate White the quickest. In the example position shown in (c), the probability of moving the Black king is 0.5, 0.25, and 0.8 with the RCT, non-RCT$_1$, and non-RCT$_2$ policies respectively.

The maze experiments demonstrate that our unified AR model can estimate interventions, partial interventions, and conditional interventions using Equations 3, 4, and 6, respectively. In contrast, a traditional offline reinforcement learning (RL) model fails to capture all three causal effects. In the chess experiments, we highlight the effectiveness of our method in estimating effects via Monte Carlo approximation using Equation 5 and its robustness to distribution shifts in the test data. Finally, the `PeerRead` setting demonstrates that the AR model can estimate effects in high-dimensional confounder scenarios and leverage pre-trained language models to improve text-based analysis. These diverse settings enable a comprehensive evaluation of the effectiveness and robustness of our approach for CI under variable-length action sequences and high-dimensional confounders such as text. Moreover, our framework is also competitive with existing methods on benchmark treatment effect estimation tasks, as shown in Appendix E.

All AR models are trained using a vanilla transformer (Vaswani et al., 2017) unless otherwise specified. Additional details on the model architecture and training process can be found in Appendix A.[1]

## 6.1 Maze navigation experiments

In this experiment, we show that a unified AR model can estimate multiple types of causal queries using Equations 3, 4, and 6. Compared to a baseline offline RL model, our approach offers greater flexibility for causal inference tasks.

We generate a synthetic maze dataset to analyze the causal effect of traversing different paths. The goal is to determine the distance to the exit after following a given path. The confounding variable $X$ represents the starting position, the action $A$ is a sequence of moves, and the outcome $Y$ denotes the distance from the final position to the exit.[2] We evaluate potential outcomes when intervening on a complete path (cf. Figure 2a) or partial path (cf. Figure 2b). The obstacle positions are fixed but are not known to the model.

The end position is fixed at the bottom-right corner, while the starting position is randomly selected from the open spaces with a probability proportional to its distance from the endpoint. Moves in the path are determined by selecting a direction based on the current position. Let the current square be in the $i$th row from the top and the $j$th column from the left. The next move is chosen according to the probabilities:

$$p_{\text{up}} = p_{\text{left}} = 0.1, \quad p_{\text{right}} = \frac{0.8i}{i+j}, \quad p_{\text{down}} = \frac{0.8j}{i+j}.$$

---

[1]Code is available at `https://github.com/jiwoongim/Deep-Autoregressive-Models-as-Causal-Inference-Engines`.

[2]$Y$ is the shortest possible distance in the maze while avoiding obstacles, not necessarily the Hamming distance.

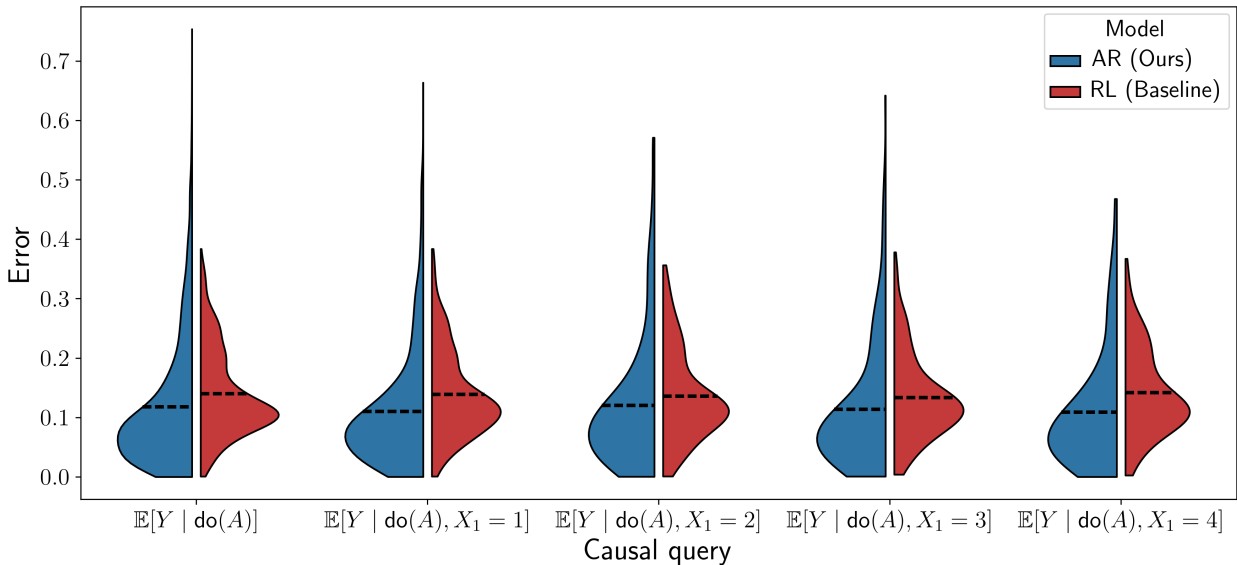

Figure 3: Error distribution of potential outcome estimates for our AR model (blue) and the offline RL baseline (red). We compute the effect of intervening on the complete path with and without conditioning on the starting row $X_1$. The plot depicts the distribution of errors across all 4096 possible paths of length six, while the black dashed line represents the mean error. The AR and RL models exhibit comparable performance across all settings, with our AR model performing marginally better.

This policy encourages paths to move towards the bottom-right corner. All actions are possible at any position, however moves that would collide with obstacles or walls in the maze are treated as no action. Paths are fixed to contain exactly six moves.

We train an AR model on sequencified data and use a deep Q-learning (DQL) model as an offline reinforcement learning (RL) baseline. The AR model is given only the starting position and must infer the effect of each move along the path. The DQL model follows the standard RL framework, where the current position is known after each move. In Appendix B, we vary the dimensionality of the maze and the length of the path to demonstrate the scalability of our framework in terms of the input and action dimension.

### 6.1.1 Causal inference using sequential actions

We estimate potential outcomes for all paths of length six in a $4 \times 4$ maze using Equation 3. Additionally, we compute potential outcomes conditioned on the starting row (from the top of the maze) $X_1 \in \{1, 2, 3, 4\}$ using Equation 6. Ground truth values are computed using the corresponding equations with the outcome outcome $\mathbb{E}_Y[Y \mid a, x]$ equal to the true number of additional moves required to reach the end of the maze after starting in position $x$ and taking path $a$. For the RL method, we predict the potential outcome as the $q$-value after taking the final action in the intervention.

Figure 3 presents the error distribution for potential outcome estimates across all paths. In all settings, both models produce estimates that closely match the ground truth. Our AR model performs comparably to the offline RL baseline in terms of mean estimation error. The AR error distribution exhibits a large tail, which can be attributed to differences in training setups. It is worth emphasizing that the RL model knows the exact position after each move in the path and therefore only needs to predict one step ahead each time, whereas our AR framework only knows the starting position and has to predict six future steps. As expected, the extra knowledge available to the RL model lowers the variance. However, even without this knowledge, our model achieves slightly lower overall error. These results demonstrate that our approach can accurately predict both intervention and conditional intervention queries.

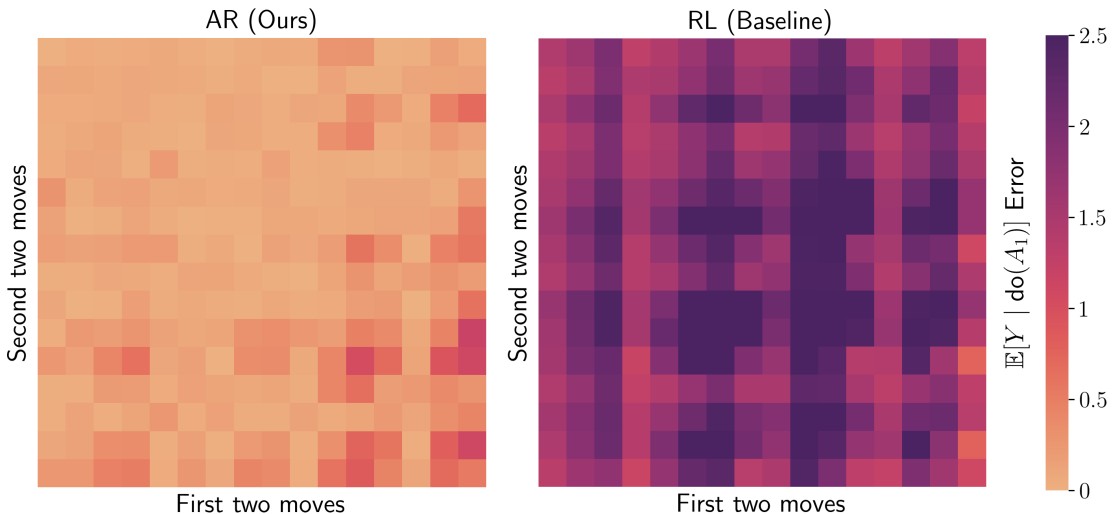

Figure 4: Heatmap illustrating the error distribution of potential outcome estimates when intervening on the first four moves in the path. Each row represents all possible actions for the first two moves, while each column represents all possible subsequent two moves. Our AR model accurately estimates potential outcomes for all four-move sequences by learning $p(\mathbf{a}_2 \mid \mathbf{a}_1, \mathbf{x})$. The offline RL baseline exhibits significant errors across nearly all interventions.

### 6.1.2 Causal inference using partial actions

In addition to computing complete interventions, we can also intervene on partial actions. To illustrate this scenario, we estimate potential outcomes when intervening on the initial moves in a path. Specifically, we decompose $A = A_1 \oplus A_2$, where $A_1$ represents the first four moves and $A_2$ the remaining two moves, and compute $p(Y \mid \mathsf{do}(A_1 = \mathbf{a}_1))$. Ground truth potential outcomes are calculated using Equation 4 by marginalizing over all possible remaining paths $A_2$. For the RL method, we intervene on the first four moves, while the remaining path is determined according to the learned policy.

Figure 4 shows the error distribution for potential outcome estimates across all possible four-move interventions. Our AR model accurately computes these estimates using Equation 4. In contrast, the offline RL model does not learn the distribution $p(\mathbf{a}_2 \mid \mathbf{a}_1, \mathbf{x})$ but instead optimizes for the best policy. As a result, it fails to predict partial interventions without modifications, such as discarding information about $A_2$ during training. This highlights the greater flexibility of our AR framework, which allows interventions on any subset of initial moves with a single model. These experiments demonstrate that a *single* AR model can accurately estimate interventional, partial interventional, and conditional interventional distributions.

### 6.2 Chess endgame experiments

In this section, we evaluate the performance of our AR model with Monte Carlo sampling, following Equation 5, and assess its robustness to distribution shifts. To explore these aspects in a more complex two-player setting, we use a synthetic chess dataset featuring king vs. king-rook endgames, where White moves first and Black holds the rook. We demonstrate that our AR model can accurately compute causal effects and identify optimal action sequences by comparing potential outcome estimates. Additionally, we leverage Monte Carlo sampling to refine estimates when only partial data is available and introduce a distribution shift between training and testing data to assess generalization.

To formulate our causal query, we ask: on average, across all starting positions, which pieces should Black move on the first two turns to checkmate White the fastest? Our question aims to uncover a general strategy for king vs. king-rook endgames, much like how controlling the center is a fundamental principle in the opening. More broadly, it parallels CI in scenarios with multiple initial conditions, where the objective is to identify the most effective strategy across a wide range of situations rather than an individual configuration.

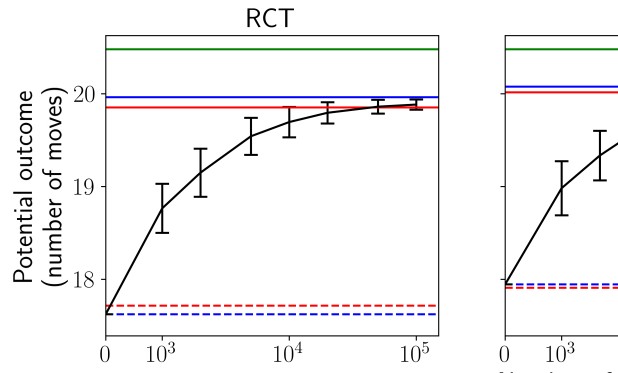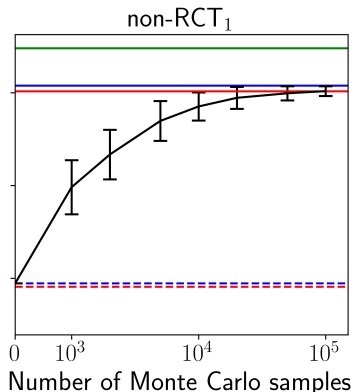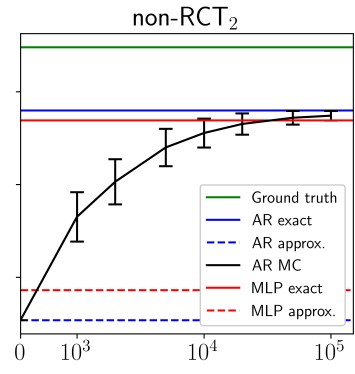

Figure 5: Potential outcome estimates for rook-king. The *exact model estimate* uses all 223,660 valid starting positions as test samples, while the *approximate* and *Monte Carlo estimates* use 1,000 randomly selected samples. Each Monte Carlo estimate was repeated 1,000 times, with error bars representing one standard deviation. By sampling from $p_\theta(X)$, the AR Monte Carlo estimate approaches the AR exact estimate. In contrast, the MLP model cannot perform sampling on $p_\theta(X)$.

Each endgame comprises a two-move chess game, potentially incomplete. The covariate $X$ is the initial piece positions. The action $A = (a_1, a_2, a_3, a_4)$ represents alternating White and Black moves. Since we consider Black's perspective, we are interested in causal quantities involving $a_2$ and $a_4$. We assume Black plays optimally after selecting which piece to maneuver, so each action only dictates whether to move the king or the rook, but not to which location. The outcome variable $y$ is the number of additional moves required to checkmate with optimal play.[3] Formally, we are interested in finding

$$(a_2^*, a_4^*) = \underset{a_2, a_4 \in \{\texttt{king}, \texttt{rook}\}}{\arg\min} \mathbb{E}_Y[Y \mid \mathsf{do}(a_2, a_4)] = \underset{a_2, a_4 \in \{\texttt{king}, \texttt{rook}\}}{\arg\min} \sum_{x, y} y \cdot p(Y = y \mid x, a_2, a_4) p(x).$$

We evaluate the ground truth using the chess engine Stockfish[4]. Figure 2c displays an example endgame.

We construct three training datasets: one Randomized Control Trial (RCT), which selects each action uniformly at random between moving the king or the rook, and two non-RCT datasets, labeled non-RCT$_1$ and non-RCT$_2$, with distinct action policies. The policy functions for non-RCT$_1$ and non-RCT$_2$ are defined as follows, where $d$ is the Hamming distance between the kings:

$$\pi_1(a_2, a_4 = \texttt{king}) = \frac{d}{16}, \quad \pi_2(a_2, a_4 = \texttt{king}) = \begin{cases} 0.8 & \text{if black king is in center } 4 \times 4 \text{ square} \\ 0.2 & \text{otherwise} \end{cases}.$$

$\pi_1$ encourages the two kings to be closer while $\pi_2$ pushes the Black king towards the edge of the board, both of which are required for checkmate. We use different RCT and non-RCT data to demonstrate robustness in the presence or absence of $X \to A$ and under varying action assignment mechanisms.

The testing dataset consists of all 223,660 valid starting positions. To assess out-of-distribution generalization, we use two distinct policies for White in the training and testing phases. White plays uniformly at random over non-optimal moves (unless no such legal moves are available) during training and plays optimally at test time. This introduces a distribution shift, which we use to evaluate generalization and robustness to new settings. Additionally, we show that our proposed AR framework produces more accurate causal estimates by using Monte Carlo sampling from $p_\theta(X)$ when given only a subset of the testing data.

We compute the ground truth potential outcomes for all actions and compare them with three different model estimates: the *exact model estimate* using the entire test dataset, the *approximate model estimate* using a subset of the test data, and the *Monte Carlo model estimate*, which also uses the same data subset

---

[3]In the event of a draw, the outcome is set to 50 due to the 50-move rule.
[4]Stockfish is available at `https://github.com/official-stockfish/Stockfish`.

but additionally generates samples from the model. The approximate and Monte Carlo estimates reflect real-world scenarios, where obtaining a large number of RCT samples is often difficult. We train a non-autoregressive multilayer perceptron (MLP) as a baseline for comparison.

Figure 5 presents our three model estimates for the AR and MLP model. As the number of samples in the Monte Carlo approximation increases, the potential outcome estimate converges to the exact estimate. Our results demonstrate that potential outcomes can be efficiently and accurately approximated with an AR model using only a fraction of the test data. Additionally, there is a gap between the AR exact model estimate and the ground truth, caused by the distribution shift in how White plays between the training and test data. The potential outcomes for the remaining actions are provided in Appendix C. By comparing all potential outcome estimates, we can answer our causal question and conclude that, aggregated across all starting positions, moving the rook twice initially is the best strategy for Black.

### 6.3 `PeerRead` experiments

We use the `PeerRead` dataset (Kang et al., 2018) to estimate causal effects in a semi-realistic setting with high-dimensional text confounders. Our results demonstrate that an AR model can leverage pre-trained language models to enhance CI in text-based settings. The dataset consists of paper draft submissions to top computer science conferences, such as NeurIPS, ICML, and ICLR, along with their acceptance or rejection decisions. We investigate the impact of including "theorems" on acceptance likelihood and evaluate how well our model captures this causal effect. Building on prior work, we focus on computational linguistics, machine learning, and artificial intelligence papers submitted between 2007 and 2017 (Veitch et al., 2020).

The covariate $X$ represents the paper's abstract text, the action $A$ is a binary variable indicating the presence of the keyword "theorem", and the outcome $Y$ is a binary variable indicating acceptance or rejection. Since real-world counterfactual outcomes are inaccessible, we follow prior methods by generating synthetic outcomes based on the action $A$ and the title buzziness $Z$ (i.e., whether the title contains "deep", "neural", "embed", or "adversarial net"). For example, $z = 1$ and $a = 1$ likely correspond to a deep learning paper that includes a theorem, while $z = 0$ and $a = 1$ may represent a theoretical machine learning paper or a deep learning paper with a theorem but without a buzzy title.

Define $\pi(z)$ as the proportion of data samples with $a_i = 1$ among those satisfying $z_i = z$. Let $\beta$ be a parameter controlling the level of confounding between title buzziness and the outcome. Following Veitch et al. (2020), we generate outcomes using the model: $Y_i \sim \text{Bernoulli}(\sigma(0.25a_i + \beta(\pi(z_i) - 0.2)))$.

Since evaluating any causal effect model requires counterfactual outcomes that are inaccessible in real-world data, we use a semi-synthetic setting, where the covariates are real-world data and the labels are generated according to patterns in the data, albeit synthetically. We demonstrate the correlation between title buzziness and the text in Appendix D.

Following the original experimental design, we report the Average Treatment Effect on the Treated (ATT),

$$\text{ATT} := p(Y = 1 \mid \mathsf{do}(A = 1),\, A = 1) - p(Y = 1 \mid \mathsf{do}(A = 0),\, A = 1),$$

across three confounding levels: low, medium, and high. A positive ATT indicates that including a theorem increases a paper's chance of acceptance. For larger values of $\beta$, the outcome becomes more correlated with title buzziness $Z$ rather than the action $A$, so the ground truth ATT is smaller.

We compare our proposed approach to a non-autoregressive MLP baseline and Causal-BERT (C-BERT) from Veitch et al. (2020).[5] Like C-BERT, we fine-tune a pre-trained BERT model on our sequencified representations for a fair comparison. Additionally, we evaluate GPT-2 (referred to as GPT), another pre-trained LLM with a comparable parameter size to BERT (Radford et al., 2019).

BERT is trained using masked language modeling (MLM) and next-sentence prediction objectives. MLM randomly masks a fraction of input tokens and trains the model to predict them. GPT is trained with a next-token prediction objective. To adapt BERT for this setting, we randomly sample subsection of the abstract during training and mask the final token to fine-tune it as a next-token prediction model.

---

[5]C-BERT learns causally sufficient embeddings: low-dimensional document representations that preserve sufficient information for causal identification and enable efficient causal effect estimation.

Table 1: `PeerRead` ATT performance across low, medium, and high confounding levels, with relative error indicated in parentheses. For our Deep Autoregressive Causal Inference Engine (DARCIE) BERT and GPT models, we quantify the uncertainty in our estimates by reporting the mean and standard error of the ATT across three trials. Entries in bold and underlined indicate best performing models for each confounding level. Overall, our DARCIE-GPT model achieves the lowest relative error, and both of our models show improvement over other methods. Training DARCIE-GPT from scratch fails to identify causal effects due to its lack of understanding of the text.

| Confounding level | Low ($\beta = 1$) | Medium ($\beta = 5$) | High ($\beta = 25$) |
|---|---|---|---|
| Ground truth | 0.062 | 0.059 | 0.028 |
| Computed biased | **0.065 (4.8%)** | 0.097 (68%) | 0.160 (470%) |
| Reported biased (Veitch et al., 2020) | 0.08 (30%) | 0.15 (150%) | 0.16 (471%) |
| MLP $\hat{\psi}^Q$ | 0.05 (20%) | 0.10 (70%) | 0.30 (970%) |
| C-BERT $\hat{\psi}^Q$ | 0.09 (45%) | **0.07 (19%)** | 0.04 (42%) |
| C-BERT $\hat{\psi}^{\mathrm{plugin}}$ | 0.10 (61%) | 0.09 (53%) | 0.05 (78%) |
| DARCIE-GPT (No pre-train) | 0.001 (98%) | 0.002 (97%) | 0.001 (96%) |
| DARCIE-BERT (Ours) | $0.039 (37\%) \pm 0.007$ | $0.039 (34\%) \pm 0.005$ | $0.016 (43\%) \pm 0.003$ |
| DARCIE-GPT (Ours) | $0.041 (34\%) \pm 0.004$ | **$0.048 (19\%) \pm 0.002$** | **$0.025 (10\%) \pm 0.002$** |

As shown in Table 1, our approach outperforms C-BERT and other benchmarks. The primary reason for the improvement is that we jointly learn representations and outcome predictions within a single model, whereas C-BERT uses additional architectural layers for multiple objective functions. To quantify the variance in effect estimation for our proposed models, we compute the mean and standard error of the ATT estimates across three trials. Note that most language models tend to have high variance due to the heavy-tailed nature of next-token prediction. Veitch et al. (2020) do not report uncertainties in their experiments.

Our proposed framework leverages the existing knowledge in pre-trained LLMs to accurately estimate causal quantities. While fine-tuning GPT yields successful results, training the model from scratch fails to identify causal effects because the model lacks prior understanding of text. We also demonstrate that our results are robust to the size of the pre-trained language model in Appendix D. By using LLMs, our approach outperforms non-autoregressive causal methods and proves more effective for a wider range of CI tasks.

## 7 Conclusion

In this work, we introduce an AR framework for CI that handles high-dimensional confounders and combinatorially large action spaces. Our proposed method, called *sequencification*, transforms data into a linear sequence of tokens based on a known causal diagram. By training a single AR model on sequencified data, we learn conditional distributions between variables in the graph. The framework enables efficient sampling and approximation of several interventional distributions in a unified manner.

We validate the effectiveness of our method for inferring causal effects across three diverse settings: maze navigation, chess endgames, and academic papers. By handling high-dimensional confounders and combinatorially large action sequences, our work extends the capabilities of CI for a wider range of applications.

Our approach has two main limitations. First, it requires the full causal graph to be known exactly, with all variables observed. A possible solution for handling unobserved values is to impute unconfounded missing data using a mask token. Second, our method supports conditioning or intervening only on variable prefixes. We leave the investigation of solutions to these limitations for future work.

## Acknowledgements

This work was supported by the Institute of Information & Communications Technology Planning & Evaluation (IITP) with a grant funded by the Ministry of Science and ICT (MSIT) of the Republic of Korea in connection with the Global AI Frontier Lab International Collaborative Research. This work was also supported by the Samsung Advanced Institute of Technology (under the project Next Generation Deep Learning: From Pattern Recognition to AI) and the National Science Foundation (under NSF Award 1922658).

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

## A  Training details

The maze experiments were conducted on a single NVIDIA Tesla T4. All models for the chess and `PeerRead` experiments were trained on a single NVIDIA GeForce RTX 3090 in four and eight hours respectively.

**Maze experiments.** The maze dataset comprises 10,000 sequencified data points. We use a vanilla transformer with 3 layers, 8 attention heads, and a hidden dimension of 64. For training, we use the Adam optimizer with a batch size of 64. The CI model is trained for 6,250 iterations, while the offline RL model is trained for 5,000 iterations.

**Chess endgame experiments.** We use a 512-dimensional vanilla transformer with 6 layers and 8 attention heads. The model is trained on a next-token prediction task using the sequencified representation. Training runs for 200 epochs with the Adam optimizer, a batch size of 4096, and a learning rate chosen to be as large as possible without overfitting.

For the training dataset, we sample 500,000 two-move chess games per dataset based on Black's policy function. The test dataset includes every game from all 223,660 legal starting positions and all four possible Black actions (king-king, king-rook, rook-king, rook-rook). We sequencify the data by assigning a unique token to each square, legal king and rook move, and outcome.

**PeerRead experiments.** We fine-tuned our models using pre-trained BERT and GPT base model checkpoints.[6] For BERT, we employed a two-phase training process similar to C-BERT. In the first phase, we trained BERT to generate abstracts, followed by a second phase where it learned to generate full sequences, including both actions and outcomes. GPT, having been pre-trained on next-token prediction, required only a single training phase. This approach ensures a gradual refinement of the generative capabilities specifically tailored to the `PeerRead` corpus. For all training phases, we trained for 100 epochs using the Adam optimizer with a batch size of 16. The learning rate was set as high as possible without overfitting.

## B  Maze extra experiments

We investigate whether the outcomes produced by our model have some interpretability. To study the predictions of our model, we reuse the synthetic maze setup in Section 6.1 with slightly modified obstacle positions. The configuration of the maze is shown in Figure 6.

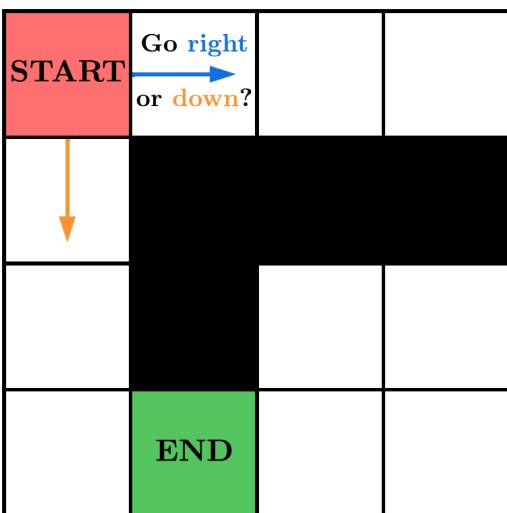

Figure 6: Illustration of the maze setup. We answer a causal question involving intervening on a partial path: is it better to go right or down?

---

[6]BERT-Base is available at `https://huggingface.co/google/bert_uncased_L-12_H-768_A-12`, and GPT is available at `https://huggingface.co/openai-community/gpt2`.

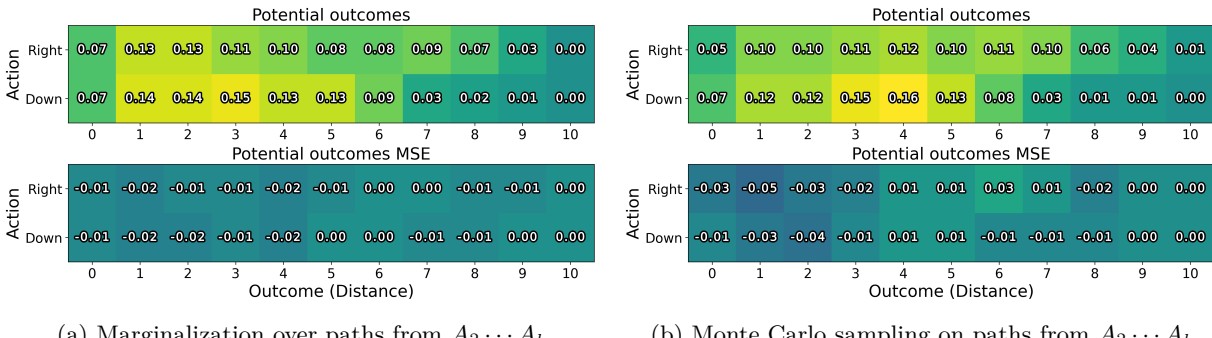

(a) Marginalization over paths from $A_2 \cdots A_k$.  (b) Monte Carlo sampling on paths from $A_2 \cdots A_k$.

Figure 7: Potential outcome MSE between ground truth and model estimates for moving right vs. down first. Both the *exact model estimate* (a) and *approximate model estimate* (b) are shown.

Following the original experimental design, the objective is to determine the distance to the exit after following a path. The confounding variable $X$ is the start and end positions, the action $A$ is the sequence of moves in a path, and the outcome $Y$ is the distance from the final position to the exit.

We consider the question: for a fixed starting state, is it better to go right or down? To answer this, we compute the potential outcome of moving right or down initially. This task is prohibitive for most non-autoregressive CI models as they treat the path as a singular variable. Our model is capable of intervening on any subsequence of actions because the conditional probability given its parents is tractable.

To compute the potential outcome, we marginalize over subsequent actions as shown in Equation 4. Since the paths may be intractable without limiting the maximum length, we cap the longest path to 16 moves as there are a total of 16 positions in the maze. To compare the exact estimates with the Monte Carlo approximations, we report both quantities derived from Equation 4 and 5, and denote them as *exact model estimate* and *approximate model estimate* respectively.

Figure 7 shows the potential outcome estimate of moving right or down initially. The distribution is more concentrated towards smaller distances for moving down and more uniform for moving right. Thus, going down is a better choice for the first move, which matches our intuition based on the maze configuration.

To demonstrate the scalability of our methodology with respect to the number of variables, we extend the maze experiments using the same setup described in Section 6.1. Specifically, we adjust the maze dimensionality and path length for fine-grained control over the covariates $X$ and actions $A$. We increase the maze from 2D (i.e., $4 \times 4$) up to 5D (i.e., $4 \times 4 \times 4 \times 4 \times 4$), while keeping the path length fixed at six. Since the starting position in $d$ dimensions is defined by $d$ coordinates, this directly controls the dimensionality of $X$. To vary the number of actions, we modify the path length from four to ten while holding the maze structure fixed at 2D. All model and training details (e.g. number of samples and number of optimization epochs) are identical to the original maze experiments.

We evaluate our model by computing the absolute error in estimating the effect of intervening on every possible path of a specified length. Figure 8 shows the distribution of these errors as we vary the maze dimensionality and path length. The average estimation error increases with both maze dimensionality and path length due to growing data complexity. The variance also increases significantly with longer paths and more gradually with higher-dimensional mazes. Nonetheless, the overall estimation accuracy remains high, with the majority of errors within 0.5 of the true effect across all configurations.

## C  Chess endgame potential outcome estimates

Table 2 compares the potential outcome values for all actions, presenting both exact model estimates and approximate model estimates. Our AR model performs similarly to the baseline across both metrics. Predictions from the model aligns with the ground truth answer to our counterfactual query: on average, moving the rook twice leads to the fastest checkmate.

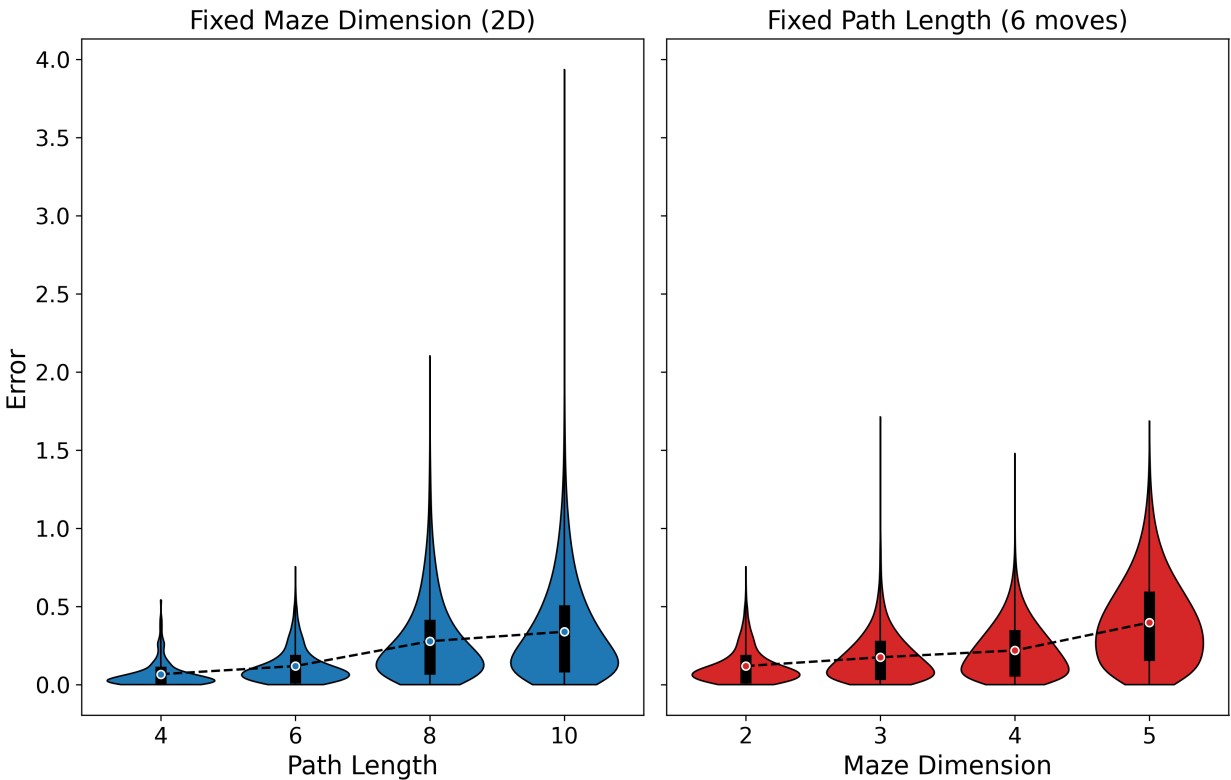

Figure 8: Distribution of effect estimation errors for all possible interventions of the given path length. The inner box indicates the lower and upper quartiles of the distribution, while the point represents the mean. The overall error increases with the input complexity (i.e. higher-dimensional mazes and longer path sequences).

Table 2: Chess endgame potential outcome estimates for all actions. The outcome represents the number of additional moves required for checkmate. Since Black aims to achieve checkmate as quickly as possible, lower values are desired.

| | | Potential outcome (Error %) | | | |
|---|---|---|---|---|---|
| | | king-king | king-rook | rook-king | rook-rook |
| Ground truth | | 22.76 | 20.18 | 20.48 | 17.27 |
| MLP exact | RCT | **22.51 (1.1%)** | **19.96 (1.1%)** | 19.85 (3.1%) | 17.08 (1.1%) |
| | non-RCT$_1$ | 22.40 (1.6%) | 19.85 (1.6%) | 20.01 (2.3%) | 17.10 (1.0%) |
| | non-RCT$_2$ | 22.24 (2.3%) | 19.90 (1.4%) | 19.69 (3.9%) | 17.14 (0.8%) |
| MLP approx. | RCT | 22.17 (2.6%) | 19.45 (3.6%) | 17.72 (13%) | 16.18 (6.3%) |
| | non-RCT$_1$ | 22.04 (3.2%) | 19.32 (4.3%) | 17.91 (13%) | 16.12 (6.6%) |
| | non-RCT$_2$ | 21.86 (4.0%) | 19.31 (4.3%) | 17.86 (13%) | 16.18 (6.3%) |
| AR exact | RCT | 22.38 (1.7%) | 19.75 (2.1%) | 19.96 (2.5%) | 16.95 (1.9%) |
| | non-RCT$_1$ | 22.48 (1.2%) | 19.87 (1.5%) | **20.08 (2.0%)** | 17.08 (1.1%) |
| | non-RCT$_2$ | 22.05 (3.1%) | 19.88 (1.5%) | 19.80 (3.3%) | **17.17 (0.6%)** |
| AR approx. | RCT | 21.90 (3.8%) | 19.19 (4.9%) | 17.62 (14%) | 15.97 (7.5%) |
| | non-RCT$_1$ | 22.03 (3.2%) | 19.33 (4.2%) | 17.95 (12%) | 16.15 (6.5%) |
| | non-RCT$_2$ | 21.79 (4.3%) | 19.29 (4.4%) | 17.54 (14%) | 16.31 (5.6%) |

In Figure 9, we present the potential outcome graphs for the remaining three actions: king-king, king-rook, and rook-rook. The model behavior on these actions closely aligns with the observations for rook-king in Figure 5. Across multiple interventions, Monte Carlo sampling improves potential outcome estimates when only a subset of the test data is available. Thus, our approach not only effectively models outcomes but also leverages Monte Carlo sampling to enhance predictions in limited data scenarios.

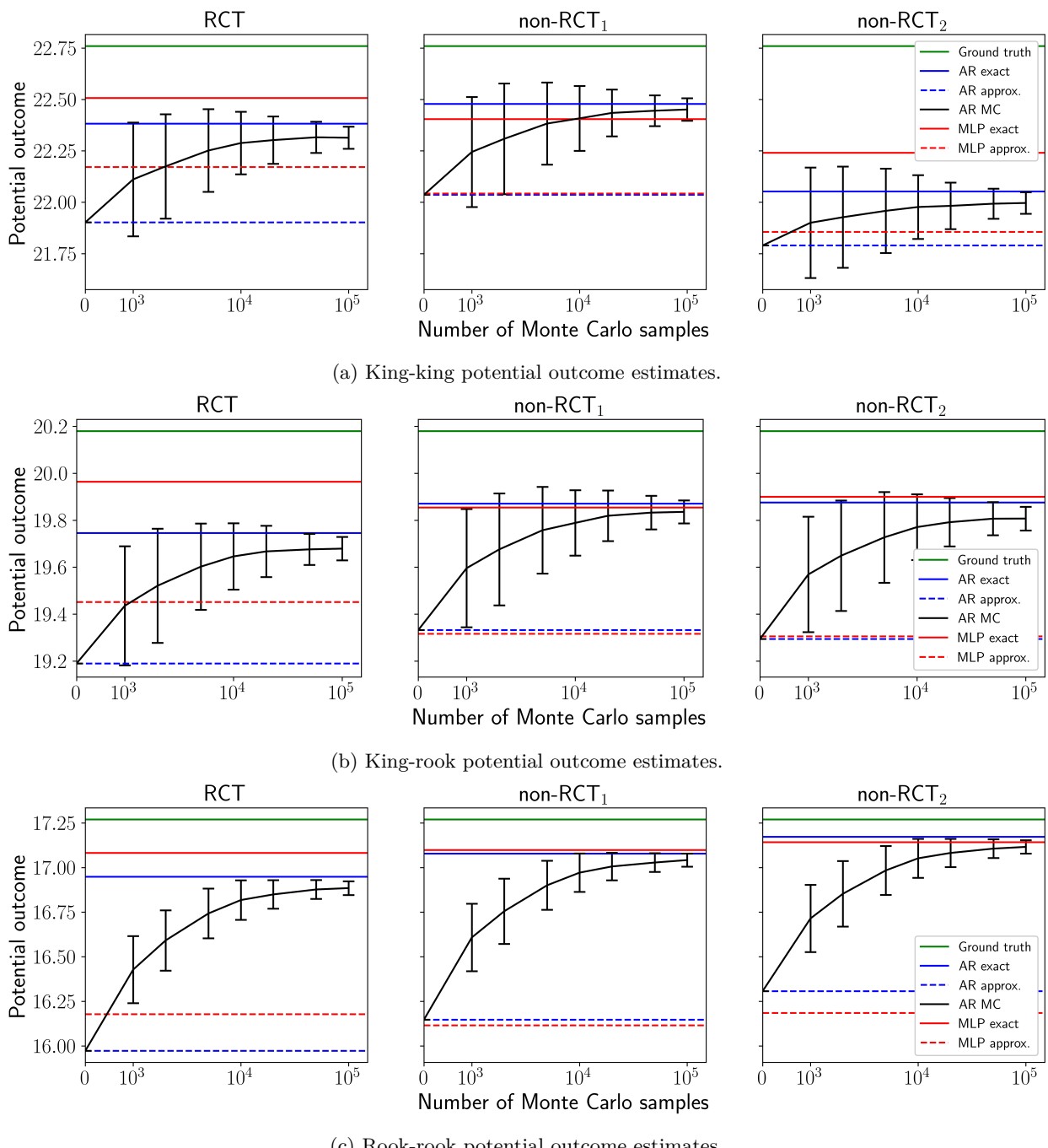

(a) King-king potential outcome estimates.

(b) King-rook potential outcome estimates.

(c) Rook-rook potential outcome estimates.

Figure 9: Potential outcome estimates for (a) king-king, (b) king-rook, and (c) rook-rook actions. Similar to the rook-king intervention, sampling from $p_\theta(X)$ enables the AR Monte Carlo estimate to approach the AR exact estimate.

# D PeerRead extra experiments

We examine the rate at which the ATT error converges to study causal effect prediction in small data cases. Using the PeerRead experimental setup, we vary the training set size from a few hundred papers to the full dataset (9305 samples). We repeat each estimation across three trials and report the mean and standard error of the relative ATT error in Figure 10. For our BERT and GPT models, the error decays gradually with the number of training samples.

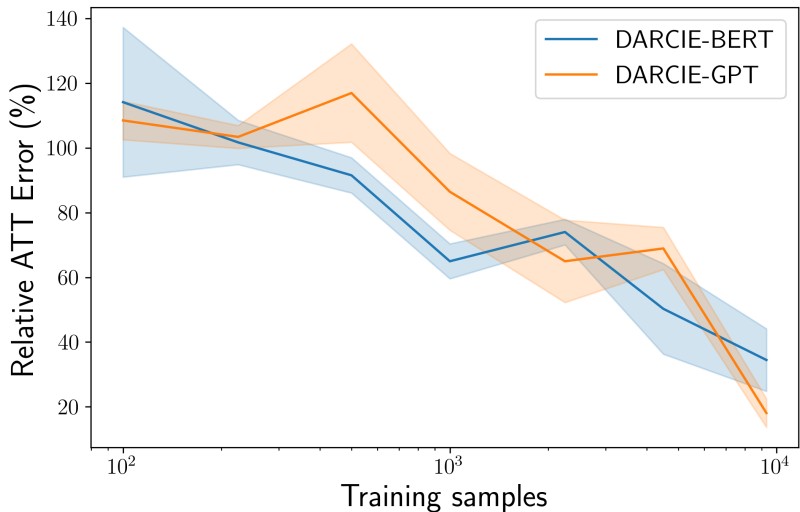

Figure 10: Mean and standard deviation of the relative ATT error while varying the training set size. The training data increases exponentially from one hundred points to the entire dataset of 9305 samples.

We also study the impact of pre-trained language model size on effect estimation accuracy using three versions of GPT. Table 3 indicates that larger models generally yield more accurate estimates.

Furthermore, we demonstrate why our model performs well when the outcome is generated from $Z$ and $A$ while the input is derived from $X$ and $A$. To quantify the correlation between title buzziness and the abstract, we train a logistic regression model to predict $Z$ from $X$. For each model, we extract the final hidden layer output as a dense vectorized representation of the abstract. As a baseline, we train a separate logistic regression model using the bag-of-words (BoW) representation of $X$. Table 4 shows a strong correlation between $X$ and $Z$, explaining the high accuracy of our potential outcome estimations.

Table 3: `PeerRead` ATT estimates for GPT small, medium, and large, with relative error shown in parentheses.

|  | # params | Low | Medium | High |
|---|---|---|---|---|
|  |  | \multicolumn{3}{c}{Confounding level} |  |
| Ground truth |  | 0.062 | 0.059 | 0.028 |
| GPT-small | 117M | 0.050 (20%) | 0.044 (25%) | 0.020 (29%) |
| GPT-medium | 345M | **0.052 (16%)** | 0.053 (10%) | 0.038 (36%) |
| GPT-large | 774M | 0.051 (18%) | **0.054 (8%)** | **0.021 (25%)** |

Table 4: Accuracy and balanced accuracy of title buzziness prediction from the abstract using logistic regression.

|  | Acc. | Balanced acc. |
|---|---|---|
| BoW | 87.95% | 73.27% |
| BERT | 88.57% | 77.91% |
| GPT2 | 89.18% | 81.00% |

To illustrate the positive predicted ATT on the `PeerRead` dataset, we compare $p_\theta(Y = 1 \mid A = 1, X)$ and $p_\theta(Y = 1 \mid A = 0, X)$ predicted by GPT in Figure 11. Our model consistently favors papers containing theorems regardless of title buzziness.

## E   Ablation studies on IHDP

Although the experiments in Section 6 are designed to demonstrate the effectiveness of our AR model in scenarios involving variable-length action sequences and high-dimensional covariates, it is also competitive with existing methods on benchmark causal datasets. To illustrate this, we evaluate our model on a semi-synthetic baseline using the Infant Health and Development Program (IHDP) data. The dataset consists of a 25-dimensional covariate $X$ (6 of which are continuous features), a binary treatment $A$, and a continuous outcome $Y$. The evaluation metric is ATE prediction error.

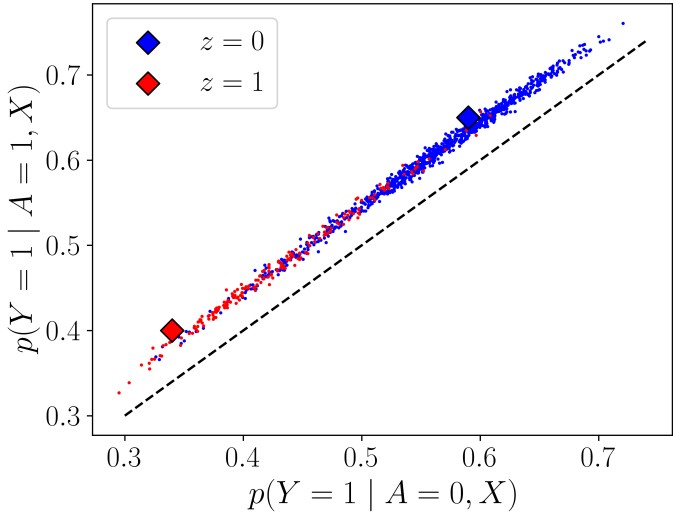

Figure 11: Conditional outcome distributions given $A = 0$ vs. $A = 1$ for medium confounding data ($\beta = 5$).

We compare performance with four other baseline models: a neural network (NN), random forest (RF), causal forest (CF), and CRNET (Zhu et al., 2024). For the AR model, we use a feedforward network embedding layer to handle continuous numerical values. We report the mean and standard error of the ATE error on the test set across 30 different random train-test splits in Table 5. Our method is competitive with common existing methods and outperforms CRNET while having lower variance.

Table 5: Evaluation of existing causal methods and our proposed AR model on the IHDP benchmark. We report the mean and standard error of the absolute and relative ATE error across 30 training trials. The best performing model is underlined and in bold.

| Model | ATE Error | ATE Relative Error (%) |
|---|---|---|
| NN | $0.1651 \pm 0.0256$ | $4.11 \pm 0.64\%$ |
| RF | $0.1171 \pm 0.0110$ | $2.92 \pm 0.27\%$ |
| CF | $\mathbf{0.1085 \pm 0.0147}$ | $\mathbf{2.70 \pm 0.37}\%$ |
| CRNET | $0.1550 \pm 0.0694$ | $3.86 \pm 1.73\%$ |
| DARCIE (Ours) | $0.1104 \pm 0.0118$ | $2.75 \pm 0.29\%$ |

