# OpenReview forum: "Deep Autoregressive Models as Causal Inference Engines"
_TMLR — Accepted by TMLR_

### Review · Reviewer_Fjad · 2025-04-13

**Summary Of Contributions:**

The paper proposes a novel causal inference (CI) framework leveraging autoregressive (AR) models—specifically language models—to estimate causal effects from data transformed into token sequences via sequencification. The approach supports high-dimensional confounders and combinatorially large action spaces, and uses a single trained model to answer multiple types of causal queries, such as interventional and conditional distributions. The framework is validated on synthetic maze navigation and chess endgame tasks, as well as the real-world PeerRead dataset.

**Audience:**

Yes

**Claims And Evidence:**

No

**Requested Changes:**

See weakness. In particular, the assumptions, motivations, the specific task the method deals with, and related works should be clearly stated and well-structured. More benchmarks and baseline methods should be provided with motivations discussed.

**Strengths And Weaknesses:**

**Strength**
1. The idea of leveraging language models to model the (conditional) distribution of the underlying DAG which features a sequential nature (through causal ordering) is interesting and novel.
2. Empirical experiments showed the effectiveness of the proposed method.

**Weakness**
1. It is not well-written and structured as a paper in the field of causal inference.
      - The introduction is too brief and merely an extension of the abstract, making it hard to follow for the general audience. For example, it does not specifically provide the causal inference task this paper is focused on (e.g. interventional distribution approximation, (individual/average) treatment effect estimation, counterfactual prediction, etc.), the motivation and practical application of the task, general ideas of AR models, why and how it can be related to your task.
      - Sections 2 and 3 are not extensive which lack the discussions of many recent papers. For example, there is already existing literature tackling high-dimensional interventional distribution approximation [1], heterogeneous treatment effect estimation with sequential/temporal treatments [2,3], and high-dimensional heterogeneous dose-response curves estimation [4,5,6]. The authors should discuss how their method and task are related to/deviated from them. Moreover, what does it mean by a causal engine? The definition is not provided. Also, why is there a $y\cdot$ in Equation 1?
      - The basic assumptions for the causal inference task are not clearly stated and discussed. For example, I assume the underlying distribution should satisfy the Markov property to be factorized sequentially. The definition of positivity condition, and how the proposed method deals with it, including the difference with existing methods, should be clearly discussed rather than briefly mentioned in Section 2. The problem setting can be defined more rigorously, c.f. Veitch et al. 2020.
      - While the description of the method is mostly clear, could you explain in more detail why the $⟨start_i⟩$ token at the start of each variable $V_i$ during sequencification helps obtain good estimates under random topological orderings? Could you provide a concrete example or pseudo code?

2. As far as I understand, all the interventional distributions are generated by conditional distributions using the valid adjustment set $X$. Therefore, any method that models the conditional distributions can serve as a "causal engine" here. While I acknowledge that language models can be powerful in modeling such distributions, it would be nice if the author could provide an illustration of the embedding of the tokens on a toy example. For example, how is the sequence of tokens of [Y] similar or different given a fixed X and different actions A?

3. Benchmarks and baseline methods.
   - I think Veitch et al. 2020 is the most relevant literature to the current paper. Therefore, I expect the authors to conduct more extensive comparisons, e.g. including the baseline method C-ATM, and the benchmark dataset Reddit. What is the computational burden compared to Veitch et al. 2020? Also, DARCIE seems to be the name of the proposed method, what does it mean?
   - I am not fully convinced that the Maze and Chess datasets are suitable for evaluating causal inference methods. The potential outcome of an interventional move is deterministic rather than probabilistic and the problems are originally from reinforcement learning (RL) and Markov decision processes. I wonder if other literature on causal inference uses this kind of dataset as the benchmark. I think the experiments are good examples of how causal inference ideas and methods can be applied to RL tasks, but not a proper evaluation of the proposed method on causal inference tasks. Note that the baseline method is also from RL.
   - The author should explain why they did not include common benchmark datasets on treatment effect estimation (which the method seems to be able to deal with) such as MIMIC, TCGA [5] IHDP, News [6], and the synthesis dataset approach they used. Please clearly state the specific tasks the proposed method can deal with.

4. Ablation studies. One of the key features of the proposed method is randomized topological orderings, the author could provide more analysis on how it contributes to the method's performance.

**References:**

[1] Melnychuk, Valentyn, Dennis Frauen, and Stefan Feuerriegel. "Normalizing flows for interventional density estimation." International Conference on Machine Learning. PMLR, (2023).

[2] Frauen, Dennis, Konstantin Hess, and Stefan Feuerriegel. "Model-agnostic meta-learners for estimating heterogeneous treatment effects over time." arXiv preprint arXiv:2407.05287 (2024).

[3] Kacprzyk, Krzysztof, et al. "ODE Discovery for Longitudinal Heterogeneous Treatment Effects Inference." The Twelfth International Conference on Learning Representations (2024).

[4] Zhu, Minqin, et al. "Contrastive balancing representation learning for heterogeneous dose-response curves estimation." Proceedings of the AAAI conference on artificial intelligence. Vol. 38. No. 15. (2024).

[5] Bica, Ioana, James Jordon, and Mihaela van der Schaar. "Estimating the effects of continuous-valued interventions using generative adversarial networks." Advances in Neural Information Processing Systems 33 (2020): 16434-16445.

[6] Nie, Lizhen, et al. "Vcnet and functional targeted regularization for learning causal effects of continuous treatments." arXiv preprint arXiv:2103.07861 (2021).

---

> ### Author Response · Authors · 2025-04-27
>
> We thank the reviewer for their detailed comments and suggestions.
>
> ## 1a.
> We will expand the introduction to address these concerns.
>
> (i) We will clarify that our framework focuses on approximating various interventional distributions in the presence of variable-length sequential actions, which encompasses tasks such as individual/average treatment effect estimation.
> - “Our framework is designed to handle variable-length sequential actions, combinatorially large action spaces, and high-dimensional confounders. These capabilities encompass a variety of common causal inference tasks including average treatment effect (ATE) estimation, individual treatment effect (ITE) estimation, interventional distribution approximation, etc. To the best of our knowledge, all prior work can only accommodate fixed-length covariates and actions, and most evaluate on relatively low-dimensional data.”
>
> (ii) We will motivate the framework through a practical healthcare example involving sequences of treatments.
> - “Consider a medical setting where a doctor prescribes a sequence of treatments to the patient. In this case, the number of treatments that the patient undergoes is not fixed (e.g. the doctor may assign treatments A and B as a substitute for treatment C), and the number of possible treatment sequences may grow combinatorially. For effective treatment effect estimation, we want a framework that can accomodate any arbitrary length of treatment/action sequences. To tackle this problem, we propose using neural network-based autoregressive (AR) models for causal inference.”
> - Our experiment in Sec 6.1.2 demonstrates our model’s capability for handling variable-length actions. The AR model accurately estimates interventions on the entire path (length 6) or partial path (length 4). This is analogous to the practical setting above where the number of medical treatments can change across different patients.
>
> (iii) We will better explain how data naturally admits a sequential ordering from the underlying causal DAG and how patterns in the data can be learned by AR models.
> - “Autoregressive (AR) models are a standard framework for learning conditional probability distributions and predicting values in sequential or time-series data. Neural-based AR models are widely used in applications such as language modeling for next-token prediction and text generation. We demonstrate that AR models can also be used to perform causal inference by turning observational data into a sequence following the underlying causal DAG. Although the causal structure is not explicitly provided, a neural network-based AR model can still learn the conditional probability distributions implied by the DAG. Moreover, because the AR model estimates all conditional distributions in the sequence, we can compute a variety of interventional distributions without relying on inverse probability/propensity weighting (IPW).”
>
> ## 1b.
> Thank you for pointing out these related studies. Papers [1–6] indeed employ deep-learning–based causal-inference models, and they operate on somewhat large-dimensional synthetic benchmarks. The key difference is that none of these works can easily handle a sequence of counterfactual covariates and actions with variable length - consider a counterfactual question in the chess experiment involving an arbitrarily long sequence of moves (king up, rook left, king down, …) and query for the potential win outcome. While [1-6] are restricted to fixed-dimension and fixed-length covariates and actions, we can accommodate a variable sequence in our setup because we are using a base autoregressive model.
>
> We will add a dedicated paragraph to Sec 2 and 3 that (i) summarizes these papers, (ii) clarifies where our setting overlaps with theirs, and (iii) highlights the key differences—specifically, variable vs. fixed-length covariates and actions.
>
> A causal engine is a model that can answer counterfactual or do-calculus queries. We have observed the term “engine” is used colloquially in the literature to refer to LLMs. Since we employ language models for answering causal questions, it made sense to call it a causal engine. We are happy to rephrase if the reviewers do not like the terminology. Furthermore, there is a $y$ in Equation 1 because the LHS is an expectation over $Y$.
>
> ## 1c.
> We will add the following description after Sec 3.1:
> - Our method relies on the following standard assumptions commonly used in causal inference (cf. Veitch 2020): unconfoundedness and positivity (We will add the relevant definitions in the paper). Furthermore, we assume that the causal graph is known and satisfies the Markov property (no unobserved confounding). Knowing the underlying Markovian causal graph lets us factorize the underlying data distribution in a causal-consistent order according to the equation in Sec 4.1. Unconfoundedness and positivity assumptions are necessary to compute interventional distributions.
>
> (Response continued to next comment...)

---

> ### Author Response · Authors · 2025-04-27
>
> ## 1d.
> The $⟨start_i⟩$ token serves as an explicit marker to identify each variable $V_i$​ during sequencification, regardless of the random topological order used. Since different samples may have different parent orders, $⟨start_i⟩$ ensures the model can always distinguish and condition correctly on the specific variable, even if its position shifts across sequences.
>
> For example, consider a simple binary system with three variables, $X, Y, Z$, where $X \to Y \gets Z$. Both $X < Z < Y$ and $Z < X < Y$ are valid topological orderings, and the data could be sequencified according to either.
> Suppose we have X=0, Z=1, Y=1, and we use the ordering $X < Z < Y$. The sequencified input would be $(⟨start_X⟩, 0, ⟨start_Z⟩, 1, ⟨start_Y⟩, 1)$. If we instead have $X=1, Z=0, Y=1$ and use the ordering $Z < X < Y$, the input would be $(⟨start_Z⟩, 0, ⟨start_X⟩, 1,⟨start_Y⟩, 1)$.
>
> Without the $⟨start_i⟩$ tokens, both sequences would be $(0, 1, 1)$, making it impossible to distinguish which value corresponds to which variable or which topological ordering was used.
>
> ## 2.
> We understand this question as the reviewer wants to know if the internal embedding produced by our model to arrive at the correct outcome has some interpretability. We did a quick experiment (see newly uploaded supplement) comparing the potential outcome of moving down vs. moving right a maze setup. The potential outcome is the number of additional moves required to reach the end of the maze while avoiding obstacles (e.g. walls). We include the distribution of potential outcomes from the model (see Figure 9 in uploaded supplement), where each row shows the distribution after moving right or down. Here, $X$ is fixed and we compare two actions to demonstrate how $Y$ changes for different $A$ (as what the reviewer asks in the example question). Notice that the distribution of potential outcomes after moving down is concentrated towards smaller values, which matches the intuition that moving down in the maze setup generally moves closer towards the end square.
>
> ## 3a.
> We compare our model against C-BERT rather than C-ATM, because Veitch et al. (2020) report that C-BERT achieves the best performance on the PeerRead dataset. We conduct our comparisons using PeerRead because it is a more established text benchmark. Nonetheless, we expect similar results on the Reddit dataset as well.
>
> Our model incurs a larger computational burden because it requires computing all conditional distributions during the forward pass, whereas C-BERT learns a single text embedding. However, our training costs are low, requiring less than eight hours for PeerRead. Veitch et al. (2020) do not provide details regarding the training time of their models.
>
> DARCIE stands for Deep AutoRegressive Causal Inference Engine, which broadly describes the structure and purpose of our framework. If the reviewers find the name of our proposed model unsuitable, we are happy to change it.
>
> ## 3b.
> To the best of our knowledge, causal inference papers do not study settings with variable-length action sequences, so the types of datasets employed between our work and the literature are slightly different. It tends to be the case that RL datasets deal with action sequences, and since we want to perform causal effect estimation on such use cases, we are borrowing RL-style datasets to demonstrate the effectiveness of our methodology. If the reviewer has any suggestions of causal datasets with variable-length actions, we would be happy to conduct experiments and compare with baselines.
>
> (Response continued to next comment...)

---

> ### Author Response · Authors · 2025-04-27
>
> ## 3c.
> We did not include common causal inference benchmark datasets (such as MIMIC, IHDP) in the main paper because these data do not involve sequence of actions or relatively high-dimensional $X$. For example, the dimensionality of the IHDP covariates is less than 30. However, our framework can handle standard benchmark queries on these datasets. For example, we show our model is competitive with others, including neural network (NN), random forest (RF), causal forest (CF), and CRNet [4], on the IHDP dataset in the table below.
>
> | Model         |    ATE Error           |   ATE Relative Error (%) |
> |---------------|---------------------|-------------------------|
> | NN            |    0.165 ± 0.140         |  4.11%                   |
> | RF            |    0.117 ± 0.060      |   2.92%                   |
> | CF            |    0.109 ± 0.081      |   2.70%                   |
> | CRNET         |    0.155 ± 0.380          |   3.86%                   |
> | DARCIE (Ours)   &nbsp; &nbsp; |    0.110 ± 0.064   &nbsp; &nbsp;    |   2.75%                   |
>
> For each model, we report the mean and standard deviation across 30 different train-test splits. In particular, our model outperforms CRNet, which also exhibits high variance. We expect similar relative performance on other benchmark datasets.
> We emphasize that our model is capable of estimating outcomes when intervening on the full action or partial action, and when conditioning on a prefix of the confounders. These capabilities encompass a broad array of standard treatment effect queries in causal inference, including ATE, CATE/ITE, PEHE, etc.
>
> Our experiments were chosen to highlight (i) comparison against standard RL models for estimating potential outcomes over a sequence of actions in the maze dataset, (ii) out-of-distribution generalization in the chess setup, and (iii) the capability to use pre-trained language models in the PeerRead experiment.
>
> ## 4.
> This is an excellent suggestion from the reviewer for strengthening the message of our paper. We will add an experiment demonstrating cases when there are multiple topological orderings. We do not expect any significant change in the performance.

---

> > ### Comment · Reviewer_Fjad · 2025-05-04
> >
> > I thank the authors for their detailed reply, which addressed most of my concerns. In particular, the motivations and related works are more detailed. The illustration of the method is clear, and the additional experiments shows that the proposed method is competitive with existing methods on the common CI benchmark.

---

### Review · Reviewer_wJja · 2025-04-13

**Summary Of Contributions:**

This paper introduces a new approach to causal inference by adapting autoregressive (AR) models to estimate causal effects. The authors position this work as bridging the gap between the capabilities of modern autoregressive models and causal inference, potentially enabling causal reasoning in complex, high-dimensional settings that were previously challenging for traditional causal inference methods. They propose a method called "sequencification" that transforms data from causal directed acyclic graphs (DAGs) into sequences of tokens, enabling a unified modeling to estimate multiple causal quantities (interventional, partial interventional, and conditional interventional distributions). They conduct empirical validation across three diverse domains: maze navigation (demonstrating the versatility of the unified model), chess endgames (showcasing Monte Carlo approximation), and academic paper acceptance prediction (leveraging pre-trained language models for text-based causal inference).

**Audience:**

Yes

**Claims And Evidence:**

Yes

**Requested Changes:**

* The authors could consider investigating how robust the approach is to misspecification of the causal graph, more complex graph and much smaller data cases.
* The paper would benefit from quantifying and discussing the uncertainty in the estimation, e.g. confidence intervals of the causal effects.
* It would be helpful to provide a discussion about the method's scalability in terms of the number of variables/dimensions and relationships, which would better support the claim of handling high-dimensional input.

**Strengths And Weaknesses:**

__Strengths__

* The paper successfully connects autoregressive modeling and causal inference in an innovative way with the help of causal ordering, addressing the limitation of high-dimensional inputs in traditional causal inference methods.
* This approach unify the computation of multiple causal quantities with a single model, rather than requiring different models for different causal queries.
* They demonstrate that the pre-trained language models improve performance on text-based causal inference tasks is promising, showing how domain knowledge captured during pre-training can enhance causal estimation.

__Weakness__
* The approach assumes complete knowledge of the causal graph with all variables observed. This is a strong assumption that limits applicability in settings where the causal structure is uncertain or where unmeasured confounding exists.
* This approach would need relatively large sample to train and thus might be sensitive to real-world small data cases and large data cases where hidden confounders are more commonly presented.
* The paper does not discuss much about the estimation uncertainty of proposed approach, which is crucial for reliable decision-making based on these estimates.
* According to the Figure 3, it seems like the estimation's instability (estimation standard deviation) increases when the input becomes more complex (changes in X1), compared to the RL method (is this because of the error accumulation?).
* There is no analysis of how sensitive the approach is to misspecification of the causal graph or to either complex graph or small data cases.

__Question__
* What's the defined causal ordering in the PeerRead experiment? It seems like a general token-wise ordering instead of phrases/concepts. Could the authors give some comments on the causal ordering in text-based/other unstructured data-based causal inference, and how the proposed approach could be applied?

---

> ### Author Response · Authors · 2025-05-08
>
> We thank the reviewer for their detailed comments and suggestions
>
> ## Requested Changes
> ### 1.
> The reviewer brings up an excellent point of studying the robustness of our model under different practical constraints.
> - **Graph misspecification.** Since most methods in the causal literature can only provide very weak guarantees on the accuracy of the causal query if the underlying DAG is misspecified, we expect that the accuracy of our model to also worsen. The key question is: does it degrade in a robust or drastic manner? We plan to include an experiment studying the quality of our causal predictions across different levels of misspecification.
> - **Small data/complex graphs.** To study causal effect prediction in small data cases, we plan to use the PeerRead experiment and vary the training set size from very few samples (\~100 points) to the full dataset (\~9000 points). Along with this, we will also include the uncertainty in the estimation. We choose the PeerRead experiment for this additional study because using text-based covariates serves as a proxy for complex interactions within $X$ in the causal graph.
>
> ### 2.
> We will add a detailed discussion on the multiple sources of uncertainty in our experiments. As we have done in our response to Reviewer PG84, we will highlight that the apparent low variance of the RL baseline is expected due to extra knowledge it has. We will also quantify the variance of causal effects by adding standard errors for the queries in the maze (Figure 3) and PeerRead (Table 1) experiments. Below, we present the mean and standard error of the ATT across three trials of the PeerRead experiment for our BERT and GPT model. The number in parenthesis indicates the relative error of the mean over the three trials compared to the ground truth ATT.
>
> |                          |                  |                 Confounding Level               |                               |
> |--------------------------|-----------------------------------|--------------------------------|-------------------------------|
> |                          | Low ($\beta = 1$)                        | Medium ($\beta = 5$)                   | High ($\beta = 25$)                  |
> | Ground truth | $0.062$ | $0.059$ | $0.028$ |
> | DARCIE-BERT (Ours)        | $0.039 (37$%$) \pm 0.007$                         | $0.039 (34$%$) \pm 0.005$                 | $0.016 (43$%$) \pm 0.003$                |
> | DARCIE-GPT (Ours)   &nbsp; &nbsp;      | $0.041 (34$%$) \pm 0.004$         &nbsp; &nbsp;               | $0.048 (19$%$) \pm 0.002$     &nbsp; &nbsp;                 | $0.025 (10$%$) \pm 0.002$                  |
> | | | |
>
> (Same table included in response to Reviewer PG84)
>
> ### 3.
> This is an excellent suggestion by the reviewer. To some extent, our current experiments demonstrate the effectiveness of our AR model across varying covariate and action space sizes. For both low-dimensional cases (two dimensions in the maze and chess experiments) and high-dimensional cases (~300 dimensions in PeerRead), the performance is competitive with existing baselines.
>
> To have more fine-grained control on the dimensionality of $X$ and $A$, we plan to extend the maze experiment by varying the “dimensionality” of the maze (2D vs. 3D vs. 4D, etc.). This will give us control over the dimension of the covariate $X$, since the starting position in a $d$-dimensional maze is specified by $d$ coordinates. The dimensionality of the action space $A$ can be varied by changing the length of the path in the maze. We will add this experiment to demonstrate the scalability of our framework in terms of the input and action dimension.
>
> ## Questions
> ### 1.
> A natural causal structure for text data is the sequential order in which tokens appear. We adopt this token-wise ordering for the PeerRead experiment. This ordering also aligns with the next-token prediction capabilities of pre-trained LLMs, which our framework can leverage to enhance performance (ref. Table 1).
>
> For other unstructured data, we can leverage the ordering from the underlying causal DAG which generates the data. Since we assume to know the structure of the DAG, any topological ordering “sequencifies” the unstructured data for our framework.
>
> ## Weaknesses
> ### 4.
> The apparent lower variance observed for the RL model in Figure 3 is due to differences in the training setup between our AR framework and the RL baseline. This is the expected behavior rather than a limitation of our model. For a more detailed discussion, please refer to Response 2 to Reviewer PG84.

---

### Review · Reviewer_PG84 · 2025-04-23

**Summary Of Contributions:**

This paper introduces a unified autoregressive (AR) model for causal inference that handles high-dimensional confounders and sequential actions. It fills the gaps left by traditional causal frameworks, which typically deal only with singleton actions.

Additionally, by transforming causal data into token sequences (sequencification), a single AR model can efficiently estimate a wide range of causal effects. Despite its simple design, the model proves effective across various tasks such as maze navigation, chess endgames, and text-based causal inference.

**Audience:**

Yes

**Claims And Evidence:**

Yes

**Requested Changes:**

1. Though the paper depends on the actual DAG structure, it is based on the chain rule P(x1,x2,...xn)=p(x1)p(x2|x1)p(x3|x1,x2)p(...). From this perspective, it is worth asking whether the exact DAG is critical for performance. I suggest the authors test the robustness of their method by evaluating performance when the assumed DAG differs from the true causal structure.

2. In Figure 3, the proposed method consistently exhibits higher variance in estimation errors compared to the baseline. It would be helpful to know whether this increased variance persists in other tasks. For example, in Table 1, could the authors include confidence intervals or standard deviations to better quantify uncertainty?

3. In the last task, the probability of treatment is a function of title Z. however, the relationship between Z and X is not well demonstrated. A simple way to demonstrate this would be to report how well  X predicts  Z This would also clarify how varying the $\beta$ parameter captures discrepancy in treatment assignment mechanisms

**Strengths And Weaknesses:**

Strengths:
1. A unified and simple structure that handles multiple queries and scale to higher dimensional input space.
2. It uses sequencification and sampling, which enable flexible and efficient inference.
3. Strong empirical results demonstrating its broad use case for various tasks.



Weaknesses:
1. It requires knowledge on the actual DAG, which limits its application in real world-setting.
2. Due to its autoregressive nature, it can only intervene on the prefixes of variable sequence.
3. One causal inference challenge is the discrepancy between treatment and control space, and it is not well understood in the paper how robust this structure is to this challenge.

---

> ### Author Response · Authors · 2025-05-08
>
> We thank the reviewer for their detailed comments and suggestions.
>
> ### 1.
> The reviewer is correct that our sequencification method does not require the exact DAG structure and the sequencification is possible from the topological ordering. However, after sequencification, we still need access to the causal graph for computing causal quantities. Like most methods in the causal literature, one can only make very weak guarantees on the accuracy of the causal query if the underlying DAG is misspecified. We plan to include an experiment studying the quality of our causal predictions across different levels of misspecification.
>
> ### 2.
>
> There are multiple sources which contribute to the uncertainty across our experiments.
> - For Figure 3, the apparent low variance of the RL baseline is expected and is primarily due to extra knowledge it has: the RL model knows the exact position after each move in the path (and therefore only needs to predict one step ahead each time), whereas our AR framework only knows the starting position (and has to predict six future steps), which adds to the uncertainty.
> - As requested, we present the mean and standard error of the ATT across three trials of the PeerRead experiment (Table 1) for our BERT and GPT model in the table below. The number in parenthesis indicates the relative error of the mean over the three trials compared to the ground truth ATT. Note that most language models tend to have high variance due to the heavy-tailed nature of next-token prediction in natural language. While Vietch et al. 2020 does not report uncertainties in their experiments, we expect similar standard errors for their baseline as well.
>
> |                          |                  |                 Confounding Level               |                               |
> |--------------------------|-----------------------------------|--------------------------------|-------------------------------|
> |                          | Low ($\beta = 1$)                        | Medium ($\beta = 5$)                   | High ($\beta = 25$)                  |
> | Ground truth | $0.062$ | $0.059$ | $0.028$ |
> | DARCIE-BERT (Ours)        | $0.039 (37$%$) \pm 0.007$                         | $0.039 (34$%$) \pm 0.005$                 | $0.016 (43$%$) \pm 0.003$                |
> | DARCIE-GPT (Ours)   &nbsp; &nbsp;      | $0.041 (34$%$) \pm 0.004$         &nbsp; &nbsp;               | $0.048 (19$%$) \pm 0.002$     &nbsp; &nbsp;                 | $0.025 (10$%$) \pm 0.002$                  |
> | | | |
>
> We will make this discussion clear in the paper.
>
> ### 3.
> We use the exact setup for PeerRead to closely compare with an existing baseline (Vietch et al. 2020). While Vietch et al. mention in their work the relationship between $X$ and $Z$, curiously, they do not have any explicit experiments supporting their claim. However, we demonstrate the relationship by training a logistic regression model to predict $Z$ from $X$ in Appendix C. Our results show there is a strong correlation between the two variables.
>
> $\beta$ controls the confounding level when generating synthetic outcomes. When $\beta$ increases, there is a stronger correlation between $Z$ and $Y$. Table 1 demonstrates that our model is robust to varying this correlation. The treatments are not being synthetically generated and are instead taken from the real-world dataset.

---

### Decision · Action_Editor_cJpk · 2025-06-09

**Recommendation:** Accept as is

**Additional Comments:**

This paper proposes a novel AR-based method to address the causal inference problem with high-dimensional confounders. This enables more downstream applications with high dimensional confounders and reliable decision making. The proposed method is demonstrated effective in various application scenarios, further showing the potential and the proposed method. The reviewers had some concerns on the technical clarity, and the concerns were all resolved in the revision. I recommend acceptance of this paper.

**Audience:**

Yes

**Audience Explanation:**

The findings of this paper is valuable to both researchers and practitioners.

**Claims And Evidence:**

Yes

**Claims Explanation:**

The proposed method is verified by various experiments on real data.